# Manipulating Microbiota in Inflammatory Bowel Disease Treatment: Clinical and Natural Product Interventions Explored

**DOI:** 10.3390/ijms241311004

**Published:** 2023-07-02

**Authors:** Mengjie Zhu, Yijie Song, Yu Xu, Hongxi Xu

**Affiliations:** 1School of Pharmacy, Shanghai University of Traditional Chinese Medicine, Shanghai 201203, China; 18602149609@163.com (M.Z.); songyijie0522@163.com (Y.S.); 2Shuguang Hospital, Shanghai University of Traditional Chinese Medicine, Shanghai 201203, China

**Keywords:** inflammatory bowel disease, gut microbiome, immunity, metabolites, synthetic drugs, natural products

## Abstract

Inflammatory bowel disease (IBD) is a complex multifactorial chronic inflammatory disease, that includes Crohn’s disease (CD) and ulcerative colitis (UC), having progressively increasing global incidence. Disturbed intestinal flora has been highlighted as an important feature of IBD and offers promising strategies for IBD remedies. A brief overview of the variations occurring in intestinal flora during IBD is presented, and the role of the gut microbiota in intestinal barrier maintenance, immune and metabolic regulation, and the absorption and supply of nutrients is reviewed. More importantly, we review drug research on gut microbiota in the past ten years, including research on clinical and natural drugs, as well as adjuvant therapies, such as Fecal Microbiota Transplantation and probiotic supplements. We also summarize the interventions and mechanisms of these drugs on gut microbiota.

## 1. Introduction

Inflammatory bowel disease (IBD), increasingly incident worldwide, is a chronic inflammatory disorder of the gastrointestinal tract, mainly including Crohn’s disease (CD) and ulcerative colitis (UC) [1]. IBD patients commonly suffer from various intestinal symptoms, such as abdominal pain, diarrhea, haematochezia and vomiting, which lead to malnutrition and weight loss [2]. IBD can induce other complications, such as arthritis, cholangitis, urinary tract infections [3], and prostatitis [4]. In addition, IBD can cause stress and psychological problems in patients, resulting in decreased libido, sexual dysfunction [5], anxiety, fatigue, and even depression [6], severely impacting the quality of life of IBD patients. It is worth noting that the incidence and prevalence of IBD is escalating globally year-on-year [1]. Unfortunately, the pathogenesis of IBD remains unclear, and its etiology is believed to be triggered by multifactorial factors, such as genetics, immunity, gut microbiota, and environment, amongst others [7]. Over recent years, the gut microbiota has been recognized as a cause and consequence of IBD and has attracted much attention in IBD pathogenesis research and biological therapies. As part of the intestinal physicochemical barrier, the gut microbiota has co-evolved with the host’s intestinal environment and is involved in intestinal maintenance, immune homeostasis, metabolism balance and nutrient supplementation [8]. Alterations in the composition of the gut microbiota may contribute to a healthy or pathological gut environment, although many findings on the intestinal flora of patients with and without IBD are not uniform or standardized. Studies have indicated differences in the compositions of the gut microbiota between healthy individuals and patients with IBD, specifically in regard to the richness and diversity of specific bacterial taxa [9]. The expansion of pro-inflammatory microorganisms, including *Ruminococcus gnavus* and *Escherichia coli*, and the reduction in anti-inflammatory microorganisms, such as *Bacteroidetes*, *Lachnospiraceae*, and *Faecalibacterium prausnitzii*, are associated with the progression of IBD [10].

In the past decade, there has been an explosive increase in the number of articles published on the microbiome in IBD [11], However, there is a lack of review on these gut microbiota and drug effects in IBD. In this review, we summarize the role of gut microbiota in the occurrence and development of IBD and provide the latest information on the treatment of IBD by means of altering the structure of gut microbiota.

## 2. Gut Microbiota Participate in the Progression of IBD

### 2.1. Interactions between Gut Microbes and Barrier

The intestinal mucosal barrier, which is the main defensive barrier against potentially harmful substances and infectious agents for the host, has been shown to closely interact with the gut microbiota [12]. The mucus provides nutrient and attachment sites for microorganisms and is associated with the production of antimicrobial mediators, including antimicrobial peptides (APMs), immunoglobulins (Ig), macrophages, etc. (Figure 1) [13]. The intestinal mucosal barrier also accommodates microbiota-derived enterotoxins, barriers, and microbial pathogen-associated molecular patterns (PAMPs), including lipopolysaccharides (LPS), peptidoglycan, muramic acid, flagellin, bacterial DNA, and double-stranded RNA of viruses (Figure 1) [14]. Meanwhile, commensal microbes can increase the concentration of mucus by promoting bacteriocin production and inhibiting pathogenic bacterial survival. For instance, the increased growth of *Akkermansia muciniphila* (*A. muciniphil*), a myxophilic bacterium in abundant mucus, can become a dominant bacterium [15]. Meanwhile, *A. muciniphil* can also upregulate the numbers of goblet cells and mucin families of the intestinal epithelium against IBD [16]. However, the defective colonic mucus layer can aggravate the invasion of pathogens and commensal-induced inflammation in IBD [17].

The continuity of the intestinal physical barrier depends on the presence of tight junctions (TJs) in intestinal epithelial cells. The expression of TJs (claudin, occludin, zonula occludes 1 (ZO-1) and cingulin) [12] is decreased in IBD and influenced by modulation of the gut microbiota. *Bifidobacteria*, a typical phylum of beneficial bacteria, has been used as an indicator species of IBD exacerbation or recovery for drug effectiveness tests [18], and can increase the secretion TJs from intestinal cells [19,20], subsequently improving the symptoms of shortened intestines in IBD mice [21]. In contrast to beneficial bacteria, enterotoxins produced by invading bacteria can severely damage the epithelial structure of the host. Most of the studies jointly indicate that the abundances of adherent *Escherichia coli* (AIEC), *Proteobacteria*, *Escherichia coli* (*E. coli*), etc., were often increased in IBD, indicating that these bacteria are possible causative agents of IBD [22]. *E. coli* at a high level in both CD and UC patients [23] can be detected in intestinal epithelial cells (IECs) [24]. AIEC, in particular, has been found to be involved in the early stages of IBD development [25]. Meanwhile, *E. coli* releases colibactin to damage DNA in IECs, which might be a possible reason that IBD patients are more likely to suffer from colorectal cancer (CRC) [26]. Toxin-bearing strains, or enterotoxigenic *B. fragilis*, cause acute and chronic intestinal disease in children and adults. For example, *Enterotoxigenic B. fragilis* damages the colonic epithelial barrier by inducing cleavage of the zonula adherent protein E-cadherin and initiating cellular signaling responses characterized by inflammation and c-Myc-dependent pro-oncogenic hyperproliferation [27]. In conclusion, the intestinal barrier function of IBD is tightly associated with the microbiota.

### 2.2. Interactions between Gut Microbes and Immune Cells

Typically, pathogens follow mucus to be captured and digested by dendritic cells and subsequently mediate more immune responses [13] (Figure 2). The gut mucosa harbors a substantial population of macrophages [28], which serve as key players in preserving gut homeostasis by responding to signals originating from the microbiota [29]. They exert significant effects on IBD by modulating pro-inflammatory (M1) or anti-inflammatory (M2) phenotypic polarization, depending on different environmental cues [30] (Figure 2). Macrophages are a double-edged sword, as their excessive activation can lead to inflammatory activation mediated by microbiota through the secretion of LPS [31]. Studies have shown that changes in gut microbiota mediated by macrophage dysfunction result in higher susceptibility to IBD [32,33]. The components of the local tissue microenvironment, such as cytokines, microbiota, microbial products, and other immune and stromal cells, more or less determine the macrophage response [28]. The damaged intestinal epithelium in IBD increases invasion of pathogenic and harmful bacteria, namely, “translocation” [34], recruiting a large number of macrophages, secreting Interleukin-1 (IL-1), IL-6, IL-12, IL-23 and tumor necrosis factor (TNF), and producing reactive oxygen species (ROS) (Figure 2) [35]. The release of cytokines and chemokines from macrophages stimulates the adaptive immune response (Figure 2), which forms many classical inflammatory pathways, including the PI3K-Akt, JAK-STAT [36] and NF-κB pathways (Figure 2).

B cells and T cells are also involved in the response to signals from microbe-induced immunity activities (Figure 2) [37]. B cells can secrete immunoglobulins (Ig) to bind intestinal *Proteobacteria*, subsequently limiting bacterial translocation to reduce inflammatory symptoms [38]. Anti-inflammatory regulatory T cells (Tregs) and pro-inflammatory helper T cells 17 (Th17) functionally antagonize each other [39], but their balance is impaired in IBD, as shown by abundant Th17 cells in the mucosa [40]. The function of Th17/Treg cells is considered a bridge linking gut microbiota with host metabolic disorders and a dependent manner for gut microbiota to ameliorate IBD [41]. *Lactobacillus* can mediate the activity of regulatory T cells to ameliorate the inflammation of IBD [42,43]. The exacerbation of IBD is always accompanied by a decreased abundance of lactic acid bacteria (LAB), which may cause intestinal acid–base disturbance [44]. *Lactobacillus* can digest host carbohydrates to produce lactate, which plays an important role in regulating intestinal Ph and the release of inflammatory factors [45].

### 2.3. The Microbiota-Derived Metabolites Involved in IBD

#### 2.3.1. Short-Chain Fatty Acids (SCFAs)

Dietary fiber or other nondigestible carbohydrates are digested by intestinal commensal bacteria into SCFAs [46], which are essential for intestinal integrity, by regulating luminal pH, promoting mucus production, providing fuel for epithelial cells and enhancing mucosal immune function [47]. The three most abundant microbiota-derived SCFAs are acetate, propionate and butyrate (at a ratio of approximately 3:1:1) [48]. Studies have noted that sodium acetate and sodium propionate have been demonstrated to exert inhibitory effects on the pathogenicity of enterohemorrhagic *Escherichia coli* [49]. A higher concentration of butyrate salts in the intestinal lumen can effectively counteract the adhesion and colonization of *Listeria monocytogenes*, resulting in a significant reduction in infection [50]. Butyrate has the ability to promote MUC2 (mucin 2) expression, leading to the restoration of the mucus barrier. Additionally, it can facilitate M2 macrophage polarization, further contributing to the repair process [51]. Sodium butyrate is a potent inhibitor of LPS-induced NF-κB, p65 and AKT signaling, inhibiting inflammation in vitro. When necessary, butyric acid can also be used as a carbon source to provide energy for the host through the β-oxidation of hydroxymethylglutaryl CoA [52]. Thus, SCFA-producing bacteria play a vital role in intestinal metabolism. Unfortunately, in the intestinal tract of patients with inflammatory bowel disease (IBD), there is consistently reduced abundance of SCFA-producing bacteria, including *Roseburia*, *Faecalibacterium*, *Prevotella 9* and *Coprococcus*, according to Jun Hu, and the decrease in SCFA-producing bacteria is accompanied by an increase in the *Escherichia-Shigella*, which is a characteristic of IBD [53]. This is consistent with the trend of serum inflammatory markers in IBD patients. They may be attributed to the phenomenon of “microbial cross-feeding”, where one microbe utilizes the end products of another microbe’s metabolism [54]. The pathogenic bacteria in IBD, such as AIEC, degrade SCFAs to counteract their anti-inflammatory effects [55], ultimately leading to immune dysregulation in the intestinal tract of IBD.

#### 2.3.2. Bile Acids (BAs)

A small fraction of free bile acids is reabsorbed directly in the gut, and a large fraction of conjugated bile acids (~95%) is absorbed in the terminal ileum [56]. During the course of circulation, primary BAs undergo various bio-transformations in the gut. Bound BAs can be hydrolyzed by bile salt hydrolase (BSH), releasing glycine or taurine and leaving free Bas. Microbiota, including *Firmicutes*, *Clostridia*, *Enterococci*, *Listeria* and *Lactobacilli*, as well as *Actinobacteria* and *Bifidobacteria*, can generate BSH to participate in BA metabolism [57]. Bile acids have direct toxic effects on bacteria through membrane damage and other effects to modulate the structure of the gut microbial community [58]. Among IBD patients, dysbiosis leads to a lack of secondary bile acids (SBAs) in the gut, and the beneficial effects of SBA supplementation on intestinal inflammation have been validated in animal models [59], perhaps as a result of SBAs inhibiting Th17 cell function [60].

#### 2.3.3. Bacterial Self-Metabolites

Histamine, which is responsible for abdominal pain in IBD patients, is mainly produced by *Klebsiella aerogenes* with high abundance in the faecal microbiota of IBD patients [61]. Increased abundance of histamine inhibits the expression of tight junction and MUC2 proteins, reduces the level of intestinal autophagy and disrupts the function of colonic goblet cells in secreting mucus, leading to defects in the intestinal mucosal barrier [15]. *Desulfovibrio*, a major type of sulfate-reducing bacteria (SRB), can trigger sulfide action resulting in frequent defecation, weight loss and increased intestinal permeability [62]. Therefore, UC patients usually have high hydrogen sulfide concentrations in the intestines. Self-metabolites of bacteria, such as colibactin and indoleamine, have DNA-damaging effects on epithelial cells and confer an increased risk of CRC [63]. The lack of tryptophan (or the increase in tryptophan metabolites in serum) has been found to worsen IBD [64]. In summary, metabolites following microbial disorders undergo a number of changes that are detrimental to IBD.

#### 2.3.4. Vitamins

Vitamin synthesis is an important metabolic function provided by the gut microbiome: *Clostridium* is implicated in the synthesis of folate, cobalamin, niacin, and thiamine [65]. *Bifidobacteria* has been implicated in folate synthesis [66], and *Bacteroides* has been implicated in the production of riboflavin, niacin, pantothenate, and pyridoxine [65]. Some intestinal bacteria are highly dependent on host-supplied vitamins [67], suggesting that vitamin deficiency can affect the growth of some bacteria or that the presence of some microbes can affect the use of host vitamins [68,69]. When dietary vitamin K is deficient, the microbial community is disordered [70], resulting in to impaired blood clotting [71]. Does this mean that vitamin K deficiency is associated with intestinal bleeding symptoms in IBD? Vitamin K1 is mainly obtained from food, and vitamin K2 is mainly synthesized by gut bacteria. For neonates or healthy people, *E. coli* can deplete the oxygen in the intestine to help other anaerobes colonize the intestine and produce vitamin K to help the intestine resist the invasion of pathogenic bacteria [72]. Moreover, Vitamin K2 can promote the abundance of short-chain fatty acids (SCFAs) in the colon and SCFA-producing genera, such as *Eubacterium_ruminantium*_group and *Faecalibaculum* [73]. In conclusion, the gut microbial host provides multiple services, including the production of important nutrients, such as amino acids, fatty acids, and vitamins [74]. Many metabolites associated with lipids, amino acids and the tricarboxylic acid cycle are significantly altered in IBD patients, demonstrating the importance of the gut microbiota [75].

## 3. Pharmacological Interventions for IBD and Their Effects on Gut Microbiota

### 3.1. Clinical Drugs

#### 3.1.1. 5-ASA

For the majority of patients with mild to severe colitis, 5-aminosalicylic acid (5-ASA) is the preferred medication, whether applied topically or orally [76]. Both oral and rectal administration of 5-ASA can alleviate symptoms, such as abdominal pain, rectal spasms, and urinary urgency in patients with inflammatory bowel disease (IBD) [77]. Furthermore, enema use of 5-ASA has a better inhibitory effect on inflammatory infiltration of intestinal epithelium [78]. Interestingly, 5-ASA did not alleviate the symptoms of dextran sulfate sodium (DSS) -induced colitis in mice with antibiotic-depleted intestinal flora, suggesting that the efficacy of 5-ASA is dependent on the intestinal flora [79] (Table 1). The importance of intestinal flora is underscored by the superiority of local action. Although 5-ASA treatment did not appear to completely reverse the disturbance of dominant bacteria in DSS-treated mice (the decrease in *Firmicutes* and the increase in *Proteobacteria* and *Bacteroidetes*), 5-ASA significantly promoted the abundance of *Bifidobacterium*, *Lachnoclostridium*, *Romboutsia* and *Anaerotruncus* and reduced the content of *Alloprevotella* and *Desulfovibrio* [79]. In addition, the concentration of 5-ASA in the mucosa is significantly correlated with the abundance of beneficial bacteria in the mucosa, but not well correlated with the abundance of bacteria in the feces [80], and it significantly inhibits *E. coli* [81]. Moreover, 5-ASA contributes to restricting the colonization of some fungi, such as ascomycetes, in the intestine, which is increased among IBD patients [82]. Although 5-ASA significantly alters the fecal metabolites of IBD patients, microbial acetyltransferases can inactivate 5-ASA [83]. Furthermore, studies found significant differences in the composition of the intestinal microbiota between a 5-ASA intolerant group and a 5-ASA tolerant group [84]. This suggests a correlation between dysbiosis and 5-ASA intolerance, so studying the relationship between gut microbiota and 5-ASA could help patients with 5-ASA tolerance.

Mesalamine (MES) is the representative drug of 5-ASA, and it is a first-line drug for the treatment and remission of moderately mild UC [97]. MES treatment reduces the relative abundance of DSS-induced *Methanobacterium* and *Vibrio* families. In particular, MES treatment reduces the abnormal abundance of DSS-induced *Methanobacteria* which characterize dysbiosis in the colitis gut flora and lead to inflammation [86]. Moreover, MES treatment can inhibit polyphosphate kinase and reduce the production of polyphosphates, which, in turn, reduces bacterial susceptibility to oxidative stress and bacterial colonization [87]. Sulfasalazine (SASP) is a precursor drug that requires bacterial azo reductase to decompose and release 5-ASA; thus, bacteria are required for its efficacy [98]. SASP, unlike 5-ASA, reversed the reduction in thick-walled bacteria and the increased abundance of *Proteobacteria* and *Bacteroidetes* in mice with colitis. SCFA-producing bacteria, such as *Lachnospiraceae* (including *Lactobacillus*) and *Rumenococci*, increased after SASP treatment. Additionally, there was an increase in lactic acid-producing bacteria, such as *Lactobacillariidae* and *Streptococcaceae* [88] (Table 1).

#### 3.1.2. Glucocorticoids

Glucocorticoids (GCs) are a powerful class of anti-inflammatory agents, but their long-term usage is associated with various side effects [99]. GC can improve the intestinal barrier in the presence of TNF, which may also be related to the regulation of intestinal flora [100]. Long-term prednisone therapy recovers α-diversity to normal levels but downregulates the relative abundance of numerous bacteria, including *Eissenbacterium* spp., *Alistites* spp., and *Clostridium* spp. Simultaneously, associated SCFAs are downregulated (such as valeric, propionic, isobutyric, and isovaleric acids), and some metabolites, such as phenyllactic acid, hydroxyphenyllactic acid, homovanillic acid and others, are upregulated [89] (Table 1). GC-induced leucine zipper (GILZ), an inducer of the anti-inflammatory effects of GC, is overexpressed in mild colonic inflammation. The fusion protein TAT-GILZ was applied with great success in preclinical models and improved the intestinal flora changes induced by DSS. Following the use of TAT-GILZ, the abundance of *Bacillus* and *Clostridium* was re-established, biodiversity increased, and the community composition more closely resembled that of non-IBD [90].

#### 3.1.3. Immunodepressants

Cyclosporine can be used to salvage or rescue refractory colitis when glucocorticoids are already out of effect or when colitis flares acutely [101]. Cyclosporine has a fairly weak effect on the composition and abundance of gut flora, but increased butyrate and acetate can be observed following cyclosporine use in healthy individuals [91] (Table 1). Azidopine significantly increased the abundance of *Bacteroidetes*, decreased the number of *Proteobacteria*, and significantly increased butyrate production during the treatment of IBD [92] (Table 1). Emerging biologic agents, represented by anti-TNF-α biologic agents, are used in the clinical management of IBD [102]. Anti-TNF-α agents not only reduce the expression of proinflammatory cytokines (such as IL-6, IL-12a, IL-17A and TNF) in the intestinal mucosa of IBD, but also alter the release of antimicrobial peptides by the intestinal flora. In addition, the abundance of *Prausnitzii* increased significantly after anti-TNF-α treatment, along with the upregulation of SCFAs [103]. The use of vedolizumab increased butyrate production but had little effect on microbial diversity [93]. Infliximab (IFX) treatment differed in microbial composition across treatment times, but the differences were small, indicating more stable bacterial and fungal compositions after IFX conditioning. *Ruminal coccus*, *Mycobacterium avium*, *Mycobacterium* spp. and *Desulfovibrio* spp. significantly increased after IFX treatment, while *Candida albicans* spp., which induce intestinal inflammation in IBD, decreased [94] (Table 1). 

#### 3.1.4. Antibiotics

Long-term use of combinations of antibiotics (amoxicillin, tetracycline, and metronidazole) result in remission rates of up to 30% in severe colitis without significant toxicity [104]. However, antibiotic use alone increases the risk of IBD recurrence, possibly due to the development of resistance [105]. The stereotype of antibiotics is to kill microorganisms, and they are therefore commonly used to deplete gut microbes and thereby perform fecal microbial transplantation (FMT), preventing infection following surgery for colitis and against recurrent *Clostridium difficile* infection [106,107,108]. After streptomycin or vancomycin treatment prior to FMT of mice with experimental colitis, enrichment of *Bacteroides*, *Parabacterium*, and *Streptococcus* spp., which included *Bifidobacterium*, *Mucispirillum*, unclassified *Clostridiaceae* I, and *Clostridium* XI, was observed. Mice subjected to FMT after metronidazole pretreatment showed a significant increase in the genus *Lactobacillus* and better ability to resist inflammation, than with streptomycin and vancomycin [95]. Ops-2071, a novel quinolone, showed low antibacterial activity against *Akkermansia muciniphila* and high antibacterial activity against other intestinal bacteria. Therefore, ops-2071 rapidly and significantly increased the proportion of *Akkermansia muciniphila* in the feces of normal rats and the production of SCFA [96] (Table 1).

### 3.2. Natural Products

Surveys have shown that natural products are more often prescribed as complementary medicine (CAM) and that patient acceptance is rising (most surveys are from China). Among CAM therapies, the most attention has been paid to medical cannabis and curcumin in treating IBD. Both of these herbal treatments have anti-inflammatory effects. Cannabis has also been shown to increase appetite and to have potent analgesic effects that may further relieve symptoms in patients with IBD [11]. A growing number of studies have found that natural products, specifically some with low bioavailability, can be present in the gastrointestinal tract for a long time and can be metabolized by gut microbiota, and, thus, direct interaction between natural products and the intestinal flora may occur. Herbal products, as potential prebiotics, have shown their effectiveness in the regulation of gut microbiota composition and the metabolism of disease-related metabolites, such as amino acids and cholesterol in IBD [109]. The active ingredients from natural plants that can treat IBD through intestinal flora, such as polysaccharides, alkaloids and flavonoids, are listed in Table 2. These findings offer new possibilities for the innovation of drugs for the treatment of IBD and are expected to lead to the development of prebiotics or new drugs.

#### 3.2.1. Polysaccharides

Polysaccharides are poorly absorbed directly by the gut, and they require assistance from gut microbes for digestion, which confirms interactions between polysaccharides and gut microbes [132]. In fact, they can serve as a carbon source and be fermented by gut microbes to produce SCFAs [133]. Natural plant polysaccharides are potent in alleviating symptoms in mice with enteritis by modulating inflammatory factors and intestinal barrier proteins. Polysaccharides derived from *Atractylodes macrocephala*, *Scutellaria*, and *Lycium barbarum* have shown potential in inhibiting the excessive production of TNF, IL-1β, and IL-6, while also promoting the expression of mucin 2, claudin 1, ZO-1, and other relevant markers [110,111,113] (Table 2). There was an upregulation of the relative beneficial bacteria abundance after administration of the *Atractylodes* polysaccharide, such as *Lactobacillus*, *Bifidobacterium*, *Bifidobacterium*, *Lachnospiraceae*, *Ruminococcus, Lachnospiraceae* and *Faecalibacterium* (SCFA-producing bacteria), as well as a decrease in the abundance of LPS-producing *Bacteroides* [110]. Polysaccharide from *Scutellaria barbata* D. Don improved DSS-induced gut microbiota dysbiosis and downregulated the abundance of harmful bacteria closely associated with cytokines such as IL-18, IL-1β, and IL-1 [111]. Polysaccharides derived from *Scutellaria baicalensis* significantly increased the SCFA levels (e.g., acetic, propionic, and butyric acids) in DSS-treated mice, markedly increased the abundances of *Firmicutes*, *Bifidobacterium*, *Lactobacillus* and *Roseburia* and significantly suppressed the levels of *Bacteroidetes*, *Proteobacteria*, and *Staphylococci* [112]. Surprisingly, *Lycium barbarum* polysaccharides not only increased the relative abundance of *Akkermansia* and *Bifidobacterium* in the intestinal flora but also promoted the growth of *Akkermansia muciniphila* and *Bifidobacterium longum* in vitro [113]. Futo brick tea polysaccharide (FBTP) can regulate the composition and structure of intestinal flora in IBD, increasing the abundance of beneficial bacteria, such as lactic acid bacteria and *Akkermansia*, as well as increasing the levels of microbial metabolites, such as SCFAs and tryptophan [114]. FBTP is utilized by the gut microbiota of IBD participants to generate SCFAs, and fermentation broth cocultured with gut microbes exhibits anti-inflammatory activity in vitro [115]. With *Dendrobium officinale* polysaccharide ingestion, *Bacteroides*, *Lactobacillus*, and *Ruminococcus* increased [116]. 

#### 3.2.2. Polyphenols

Phenolic compounds exhibit potent anti-inflammatory and antioxidant activities in vitro and in vivo and are a major class of agents against IBD, mainly flavonoids, catechins, stilbenes, coumarins and phenolic acids [134]. Curcumin, a hydrophobic polyphenol extracted from the roots of turmeric, can protect mice against DSS-induced colitis by enhancing the abundance and stability of intestinal flora, upregulating the proportion of *Ruminococcaceae*, *Lachnospiraceae*, *Muribaculaceae*, and *Prevotellaceae*, and downregulating the relative abundance of *Helicobacteraceae*, *Desulfouvibrionaceae* and *Marinifilaceae* [117]. The beneficial effects of phytoflavonoids on the intestine are well established. As an example, total flavonoids from Liquorice can inhibit the activation of the NLRP3 inflammasome triggered by irinotecan (CPT-11) and regulate CPT-11-induced faecal metabolic disorder in mice, mainly hypoxanthine and uric acid in purine metabolism. In addition, total flavonoids from licorice increased the abundance of lactic acid bacteria and butyrate-producing bacteria such as *Roseburia* [118]. Licoflavone B remodels the intestinal microbial system by inhibiting harmful bacteria (*Escherichia coli*, etc.) and increasing beneficial bacteria (*Lactobacillus*, *Eubacterium*, etc.) [119]. Licochalcone A can not only repair intestinal TJs, but also up regulate probiotics beneficial to IBD [135]. Some studies have found that chalcone has a good inhibitory effect on bile salt hydrolases, which may be the reason why drugs change the bacterial metabolic spectrum [136]. Galangin was shown to improve the diversity of the gut microbiota, increase the levels of SCFA, and restore the abundance of *Lactobacillus* and *Butyricicoccus*, potentially explaining its protective effect against colitis [120]. Cryptotanshinone can alleviate chemotherapy-induced colitis by reducing faecal flora-associated lipid metabolism [137]. Natural polyphenols are excellent supplements for intestinal bacteria, and more polyphenols are being found in medicinal plants. 

#### 3.2.3. Alkaloids

First among the alkaloids used in the treatment of IBD is berberine, while recent studies suggest that its therapeutic effects might be related to flora. Administration of berberine to mice with colitis resulted in an increase in lactic acid-producing bacteria and a decrease in carbohydrate-hydrolysing bacteria and conditional pathogenic bacteria, followed by a reregulation of flora-mediated amino acid metabolism and biosynthesis. Furthermore, carbohydrate metabolism and glucose metabolism were improved with berberine treatment [124]. Increased serum amino acid and faecal tryptophan metabolism after berberine treatment is believed to be highly correlated with the gut flora [124,125]. Additionally, berberine may prevent IBD-associated cancer or tumorigenesis by reducing the abundance of cancer-associated bacteria [126,138]. Evodiamine has a potential therapeutic effect on IBD by modulating the *Firmicutes*/*Bacteroidetes* ratio, increasing the abundance of *L. acidophilus* and the level of acetic acid, and promoting the increase in goblet cells and secretion of antimicrobial peptides [127]. Alkaloids are also fermented by bacteria to produce SCFA, and regulate the structure of gut microbiota in various ways, It is expected to become a common pathway for this type of disease.

#### 3.2.4. Glycosides

Ginsenosides are representative glycosides for the treatment of IBD [129]. Ginsenoside RK2 restored cellular function in human intestinal epithelial THP-1 cells by inactivating the ERK/MEK pathway and reducing the release of inflammatory factors [139]. Ginsenoside Rg1 can condition intestinal flora by inducing macrophage polarization, which reduces the abundance of *Bacteroides* and *Staphylococcus* and increases the levels of flora-associated metabolites [130]. Icariin significantly reduced the proportion and activity of *Bacteroides*, the *Helicobacter pylori* family and *Turicibacter* and appreciably increased the proportion and viability of beneficial flora (*Lactobacillus*, *Lachnospiraceae*, and *Akkermansia*), ameliorating colon damage [140].

#### 3.2.5. Traditional Chinese Medicine Compound Formulas

Traditional Chinese medicine (TCM) compound formulas have unique advantages in the treatment of IBD due to their collocation of varied natural materials. For example, *Baitouweng* decoction can enhance autophagy through the PI3K-Akt-mTORC1 signalling pathway and regulate intestinal flora structure and metabolites through the IL-6/STAT3 signaling pathway and FXR and TGR5 pathways [141,142,143]. *Dahuang mudan* decoction ameliorated colitis in mice by restoring the Th17/Treg balance, recovering the α-diversity, increasing the abundance of *Firmicutes* and *Actinobacteria*, decreasing the abundance of *Proteobacteria* and *Bacteroidetes*, increasing the number of butyrate-producing bacteria (namely, *Butyricicoccus baumannii*), and restoring intestinal SCFA content [144]. *Gancao xiexin* decoction exerts therapeutic effects on IBD dependent on gut microbiota, which fails to reduce DAI scores or alleviate either colonic shortening or colonic damage in germ-free mice [145]. Other formulas include *Gegen Qinlian* decoction, *Lizhong* decoction, etc., as shown in Table 3.

### 3.3. Other Therapeutic Strategies

#### 3.3.1. Diet and Nutrition

A Western diet, characterized by high protein, high fat, high sugar and low fiber has been shown to predispose individuals to IBD, and it yields a reduction in microbial diversity and damages colonic mucus, predisposing bacteria to expansion and activity leading to the accumulation of specific immune cell populations and significantly altering the nutrient absorption function of enterocytes [154]. After UC patients received a low-fat high-fiber diet, the relative abundance of actinobacteria decreased, and the abundance of *Faecalibacterium prausnitzii* increased, along with an increased relative abundance of anti-inflammatory metabolites (such as acetate) in the feces [155]. Investigations have found that the majority of IBD patients have low vitamin D levels, and vitamin D receptor intestinal (VDR) expression is inversely correlated with the severity of inflammation in IBD patients [156,157]. The VDR pathway is a promising target for the prevention of high-fat diet-induced inflammatory bowel disease, according to O’Mahony, C. [158]. Vitamin D has a positive regulatory effect on the gut microbiota structure in IBD, increasing beneficial bacteria, such as *Roseburia*, *Alistipes*, *Parabacteroides*, and *Faecalibacterium*., while reducing pathogenic bacteria, such as *Ruminococcus gnavus*. However, the effects are transient, meaning that long-term supplementation of vitamin D alone cannot sustain this microbial structure [159].

#### 3.3.2. Fecal Microbiota Transplantation

Treatment of digestive diseases, such as severe diarrhea with juice of feces, also called “Huanglong Tang”, has been documented as early as in Zhou Hou Bei Ji Fang over 1000 years ago. FMT is a direct means of reshaping the intestinal flora, with comparable therapeutic efficacy to that of GCs and with a higher safety profile [160]. Fecal transplant patients can inherit the donor’s glycoside hydrolase genes (increasing digestion of dietary polysaccharides), butyrate biosynthesis genes, and mucin digestion genes, among others. Unfortunately, depletion of donor *Enterobacteria* in recipients tends to occur more rapidly than colonization [161]. In fact, whether drug treatment or probiotic supplementation is administered, there is also the problem that any alteration may be temporary and that once the medication is discontinued, the structure regarded as healthy may quickly be disrupted again. Therefore, post stabilization with altered intestinal structure is an issue that should also be considered when treating intestinal diseases based on gut microbes. FMT is also commonly used as a test of whether a drug targets the gut flora. Usually, fecal bacteria of mice after administration are used for FMT; for example, Wu, J. et al. transplanted the fecal microbiota of Rhein-treated mice into colitis mice to verify whether rhein perturbed gut flora and had a role in ameliorating colitis [122].

#### 3.3.3. Probiotic Supplementation

In view of the benefits of probiotics in the intestinal tract, probiotic supplements have been derived to compensate for the reduction in probiotics in IBD. A number of probiotic supplements have been proven to be effective, although the European Medicines Agency has not approved any probiotics to be marketed in pharmaceutical form. The major flora deemed to be beneficial for IBD patients include lactic acid-producing bacteria, *Bifidobacteria*, *Bacillus* and *Enterobacter*. Representative probiotic supplements of *Bifidobacterium* spp. contain *Bifidobacterium* BLa80, *Bifidobacterium longum* CECT 7894, *Bifidobacterium bifidum* BD-1, *Bifidobacterium bifidum* H3-R2, *Bifidobacterium bifidum* H3-R2, among others [162,163,164,165,166]. Representative beneficial strains of *Lactobacillaceae* include *Lactobacillus paracasei* IJH-SONE68, *Lactobacillus paracasei* L9, *Lactobacillus casei* strain Shirota, *Bifidobacterium lactis* BL-99 and others [167,168,169,170] (shown in Table 4). Probiotic supplementation can not only supplement the absence of flora in the intestine, but also improve the effective rate of drugs [171]. For example, combining *Bifidobacterium* and mesalazine not only relieves IBD, but also reduces adverse effects [171]. Therefore, rational use of the benefits of the gut flora may achieve a ‘1 + 1 > 2’ effect. 

## 4. Conclusions

In conclusion, gut microbes possess great potential, and their contribution to the treatment of IBD cannot be ignored. A stable gut microbial structure is a prerequisite for the gut to perform intricate physiological processes. Gut microbes directly contact the intestinal epithelium, influence mucus secretion and mucosal immunity, and mediate the differentiation of intestinal stem cells. Structural and metabolite changes in the gut microbiota are emerging as a new class of parameters for drug development and mechanistic studies. In this review, we scrutinized the changes in, and functions of, gut microbes in IBD and summarized the latest therapeutic approaches to IBD, including 5-ASA, immunosuppressants, glucocorticoids, polyphenols, polysaccharides and herbal combinations, and their effects on the gut microbiota. Most of these drugs reverse the changes in gut flora in IBD, which specifically increase certain beneficial (*Bifidobacteria*, *Lactobacillus*) or metabolites (SCFA, SBAS, amino acids, etc.), whilst decreasing harmful bacteria (Adherent *Escherichia coli*, *Proteobacteria*, *Escherichia coli*). Interestingly, most of the drugs that are beneficial for IBD allow beneficial microorganisms to dominate the gut and reduce harmful ones. 

The complexity of gut microbiota has increased the challenge to the mechanism research of drugs and gut microbiota, so the current mechanism research is not deep enough. Nevertheless, the mechanism, or causal relationship, between IBD and gut microbiota is also controversial. Reconstitution of the gut flora structure can reverse the exacerbation of IBD, although it is unclear whether this change is the cause or the effect. However, existing research lacks long-term tracking of gut microbiota and IBD. Since IBD is typically a chronic condition, and patients’ disease statuses and treatments can vary over time, it is challenging to establish a clear understanding of the relationship between gut microbiota and IBD or its treatment. Fortunately, advanced metagenomic and metabolomic analyses provide deep insights into the intricate relationships among gut microbes, metabolites, and hosts. Ongoing analyses of microbial systems are propelling rational development of targeted microbial drugs. 

Gradual adverse reactions to IBD drugs are commonly exposed in clinics. This defect seems to be compensated by regulating microbiota. The customized gut microbiota structure can meet the individualized requirements of patients, that is, in combination with FMT, and has been proven to be effective [160,174]. It is claimed that natural products are free from contamination, and they are often further processed by gut microbiota. Therefore, natural products possess tremendous potential in the treatment of IBD. However, many studies on natural products mediating the gut microbiota seem to be limited to investigating the outcomes of improving the gut microbiota without delving deeper. Additionally, the issue of the long-term stability of the newly established microbial structure resulting from such interventions remains unclarified. Similar limitations are observed in the field of FMT, where low rates of successful microbial engraftment and limited sustainability of therapeutic effects are common. Furthermore, there is a lack of standardization in research methods for studying gut microbiota. The complexity of the microbial community and individual variations among patients contribute to diverse responses to therapeutic interventions, and there is often insufficient sample size support. Therefore, further efforts are needed in the field of microbial therapeutics research.

## Figures and Tables

**Figure 1 ijms-24-11004-f001:**
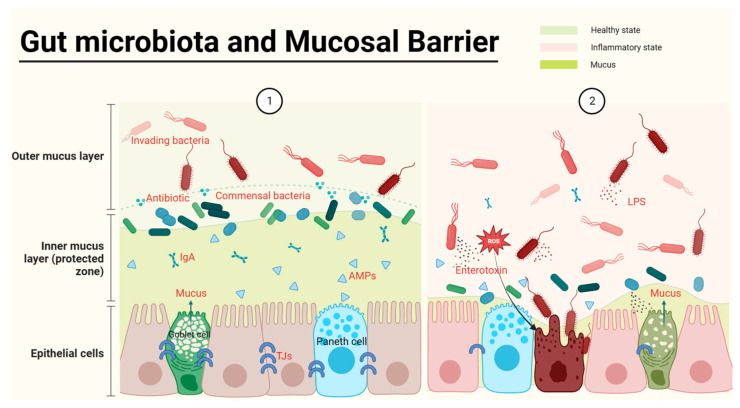
Gut microbiota and the epithelial barrier. The healthy gut (①) is full of abundant mucus and gut microbes, with commensal bacteria producing antibiotics to defend against harmful bacteria and to spatially crowd out harmful bacteria. In the inner layer of the barrier, the intestinal epithelium is tightly interconnected, and the production of antimicrobial peptides by Paneth cells as well as Ig secreted by immune cells protects the safety of the intestinal epithelium. When harmful bacteria become dominant bacteria, they consume mucus and release PAMPs, leading to impaired intestinal epithelial tightness and inflammation (②).

**Figure 2 ijms-24-11004-f002:**
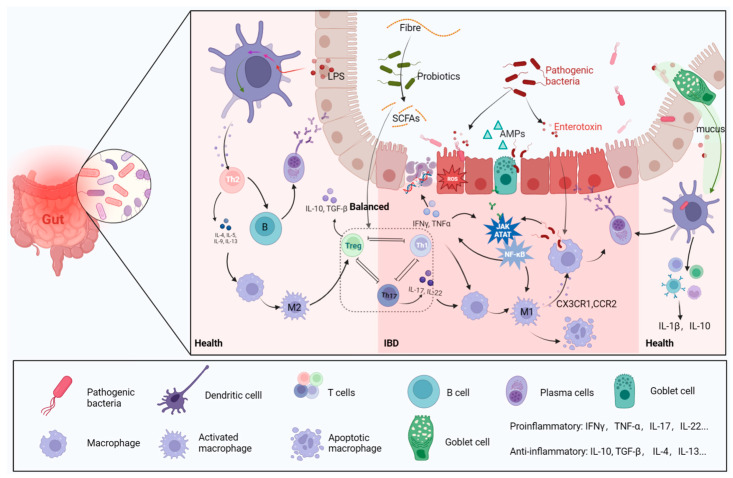
Gut microbes and immune cells. In a healthy intestinal environment, antigen-presenting cells transmit antigenic signals to T cells, stimulating immune responses. The following processes are engaged, and assisted, by beneficial bacteria: Th2 cell-mediated anti-inflammatory response, B-cell to plasma cell differentiation to produce antibodies, Th cells differentiating into Treg cells, and macrophage evolution into M2 type immune system. When the intestinal barrier is broken, harmful bacteria and their metabolites invade the intestine, which, in turn, induces the release of a large number of pro-inflammatory factors, leading to oxidative stress and damage to epithelial cell DNA. M1 macrophages and the inflammatory pathways they mediate are activated and contribute to the progression of IBD.

**Table 1 ijms-24-11004-t001:** Synthetic drugs for IBD and their effects on the gut microbiota.

Therapy	Type	Representative Drugs	Effects on Microbes	Refs
Synthetic drugs	Aminosalicylates	5-ASA	Remolding the disturbed gut microbial community structure; Increasing the number of *Ascomycetes*; Inhibiting the number of *E. coli*; Reducing fungal colonization in the gut; Improving abnormal metabolism of SBAs and indole.	[79,81,82,85]
		Mesalazine	Remodeling the abundances of intestinal microorganisms; Suppressing the abundance of *Methanobacter* spp.; Reducing the accumulation of polyPs.	[86,87]
		Sulfasalazine	Increasing the *Firmicutes*/*Bacteroides* ratio. Increasing abundance of SCFAs-producing bacteria and lactic acid-producing bacteria.	[88]
	Glucocorticoids	Prednisone	Upregulating some metabolites and altercating the structure of metabolites.	[89]
		TAT-GILC	Increasing the abundances of *Bacillus* and *Clostridium*; Improving biodiversity; Modulacing bacterial community composition.	[90]
	Immunodepressants	Cyclosporine	Increasing the production of butyrate and acetate.	[91]
		Azathioprine	Promoting butyrate production; Reducing *Proteobacteria*.	[92]
	Biologic agents	Vedolizumab	Increasing the production of butyrate.	[93]
		Infliximab	Stabilizing the intestinal flora; Increasing the abundance of beneficial bacteria; Reducing the number of *C. albicans*.	[94]
	Antibiotics	Metronidazole	Increased lactic acid bacteria.	[95]
		OPS-2071	Increasing the proportion of *A. Muciniphila*; Increasing SCFAs.	[96]

**Table 2 ijms-24-11004-t002:** The Regulatory Effects of Natural Products on the Intestinal Microbes of IBD.

Effective Compounds	Name	From	Effects on Microbiota	Refs
Polysaccharides	\	*Atractylodes macrocephalae* Koidz.	Enhancing overall richness and diversity; Reducing abundance of *Proteobacteria*; Reducing abundance of LPS-producing *Bacteroides*; Increasing abundance of *Bifidobacterial*, *Lactobacillus* and et al.	[110]
	\	*Scutellaria barbata* D. Don	Reducing the abundance of IFN-γ-, IL-1 β-, IL-6- and IL-18-associated gut flora.	[111]
	\	*Scutellaria baicalensis* Georgi	Increasing the levels of acetate, propionate and butyrate; Increasing the abundances of Firmicutes, *Bifidobacteria*, *Lactobacilli*, and *Roseburia* spp.; Suppressing the levels of *Bacteroidetes*, *Proteobacteria*, and *Staphylococci*.	[112]
	\	*Lycium barbarum* L.	Increasing the abundances of *Akkermansia* and *Bifidobacterium*.	[113]
	\	*Poria cocos* (Schw.) Wolf	Modulating gut microbiota structure to a healthy level; Increasing SCFAs; Increasing amino acid metabolism (tryptophan); Being utilized by the gut flora (producing SCFAs).	[114,115]
	\	*Dendrobium officinale* Kimura et Migo	Increasing the abundances of *Bacteroides* spp., *Ruminococcus* spp., *Akkermansia* spp, and etc.; Increasing the levels of SCFAs.	[116]
Polyphenol	Curcumin	*Curcuma longa* L.	Increasing the ratio of Ruminococcaceae, Lachnospiraceae, Muribaculaceae, and Prevotellaceae; Decreasing the relative abundance of Helicobacteraceae, Desulfouvibrionaceae and Marinifilaceae.	[117]
	\	*Glycyrrhiza uralensis* Fisch	Decreasing hypoxanthine and uric acid; Increasing abundance of *Lactobacillus* and butyrate-producing bacteria.	[118]
	Licoflavone B	*Glycyrrhiza uralensis* Fisch	Increasing abundance of the gut microbiota; Altering structure of gut microbiota.	[119]
	Galangin	*Alpinia officinarum* Hance	Increasing levels of SCFAs; Restoring the abundance of *Lactobacillus* and increased the number of *Butyrimidomonas*.	[120]
	Juglone	*Juglans regia* Linn.	Increasing the proportion of *Firmicutes* and *Bacteroides*; Increasing the abundance of *Actinomycetes*; Reducing the abundance of *Verruca* flora.	[121]
	Rhein	*Rheum palmatum* L.	Increasing Lactobacillus; Increasing the production of uricase by bacteria and then reducing uric acid; Decreasing pathogenic bacteria (e.g., Enterobacteriaceae and Turicibacter).	[122,123]
Alkaloid	Berberine	Berberine	Increasing levels of SCFAs; Increasing abundance of lactate-producing bacteria; Increasing metabolism of amino acids.	[124,125,126]
	Evodiamine	Evodiamine	Increasing *L. acidophilus* levels; Increasing production of acetate.	[127]
	Oxymatrine/total matrine	*Sophora alopecuroides* L.	Reversing of gut microbiota structure; Adjusting bile acid metabolism.	[128]
Glycosides	Ginsenoside Rg1	*Panax ginseng* C. A. Mey.	Increasing the levels of SCFAs; Restoring the abundance of *Lactobacillus* and increased the number of *Butyrimidomonas*.	[129,130]
	Baicalin	*Scutellaria baicalensis* Georgi	Increasing the abundance of butyrate-producing bacteria (e.g., *Butyricimonas* spp., *Roseburia* spp., *Subdoligranulum* spp., and *Eubacterium* spp.).	[131]

Abbreviations: IFN-γ: Interferon-γ.

**Table 3 ijms-24-11004-t003:** The effects of TCM compound formulas on gut microbiota for IBD.

Name	Compounds	Mechanism	Refs
Baitouweng Decoction	*Pulsatilla chinensis* (Bge.) Regel, *Coptis chinensis* Franch., *Phellodendron chinense* Schneid., *Coptis chinensis* Franch., *Fraxinus rhynchophylla* Hance	Promoting autophagy and anti-inflammation; Decreasing the abundance of *Bacteroidetes* and increasing the abundance of *Firmicutes* and lactic acid bacteria; Increasing metabolites of bile acid and tryptophan;	[141,143]
Dahuang Mudan Decoction	*Rheum palmatum* L., *Paeonia suffruticosa* Andr., *Prunus persica* (L.) Batsch, Na_2_SO_4_·10H_2_O, Benincasae semen	Upregulating the *Firmicutes*/*Bacteroidetes* ratio; Increasing the number of *Butyricicoccus pullicaecorum* (a butyrate-producing bacterium); Restoring Th17/Treg balance;	[144]
Gancao Xiexin Decoction	*Glycyrrhiza uralensis* Fisch., *Scutellaria baicalensis Georgi, Zingiber officinale* Rosc., *Pinellia ternata* (Thunb.) Breit., *Ziziphus jujuba* Mill., *Coptis chinensis* Franch.	Decreasing the proportion of *Proteobacteria*; *Desulfovibrio* and *Deltaproteobacteria* were enriched;	[145]
Huangqin Decoction	*Scutellaria baicalensis* Georgi, *Paeonia lactiflora* Pall., *Glycyrrhiza uralensis* Fisch., *Ziziphus jujuba* Mill.	Reducing intestinal invasion by bacteria; Upregulating amino acid metabolism; Inhibiting TLR4 pathways; Inhibiting the NOD2-dependent pathway;	[146,147]
Gegen Qinlian Decoction	*Pueraria lobata* (Willd.) Ohwi, *Scutellaria baicalensis* Georgi, *Coptis chinensis* Franch., *Glycyrrhiza uralensis* Fisch.	Increasing numbers of *Lactobacillus* and *Akkermansia*; Augmenting immunity; Protecting the intestinal barrier and upregulating ZO-1 and occludin;	[148,149]
Lizhong Decoction	*Panax ginseng* C. A. Mey., *Atractylodes macrocephala* Koidz., *Glycyrrhiza uralensis* Fisch., *Zingiber officinale* Rosc.	Regulating metabolism (such as purine metabolism, secondary bile acid biosynthesis, tryptophan metabolism, and glycerophospholipid metabolism)	[150]
Qingchang Huashi Formula	*Coptis chinensis* Franch., *Scutellaria baicalensis* Georgi, *Pulsatilla chinensis* (Bge.) Regel, *Aucklandia lappa* Decne., and et al.	Restoring the Firmicutes/Bacteroidetes ratio; Downregulating inflammation by blocking the NLRP3/IL-1β signaling pathways;	[151]
Shenling Baizhu Powder	*Dolichos lablab* L., *Atractylodes macrocephala* Koidz., *Poria cocos* (Schw.) Wolf, *Glycyrrhiza uralensis* Fisch., and et al.	Increasing tryptophan metabolism;	[152]
Huaihua Powder	*Sophora japonica* L., *Platycladus orientalis* (L.) Franco, *Schizonepeta tenuisfolia* Briq. In addition, et al.	Restoring the abundance of *Firmicutes*;	[153]

Abbreviations: NLRP3: NOD-like receptor family pyrin domain-containing protein 3; TLR4: Toll-like receptor 4; NOD2: Nucleotide-binding oligomerization domain-containing protein 2.

**Table 4 ijms-24-11004-t004:** Probiotic supplementation for IBD.

Classification	Name	Mechanism	Refs
Bifidobacteria	*Bifidobacterium bifidum* BGN4	Maintaining the intestinal barrier; Inhibiting NF-κB activates signaling molecules.	[162]
	*Bifidobacterium bifidum* H3-R2	Inhibiting of NF-κB pathways; Regulating expression of TJs; Reducing LPS-induced disruption of barrier in IEC.	[163]
	*Bifidobacterium* BLa80	Inhibiting inflammation; Promoting the growth of beneficial bacteria.	[165]
	*Bifidobacterium longum* CECT 7894	Increasing ZO-1 and Occludin expression; Increasing relative abundances of the genera *Bifidobacterium*, *Butyricicoccus* and *Clostridium* and decreased relative abundances of the genera *Enterococcus* and *Pseudomonas*; Upregulating the levels of SBAs such as a-mca, ß-MCA, LCA, CDCA, UDCA, etc.	[166]
Lactobacillus	*Lactobacillus paracasei* L9	Increasing numbers of butyrate-producing bacteria; Inhibiting IL-6/STAT3 signaling and increasing IL-6 expression in colitis.	[168]
	*Lactobacillus casei* Strain	Increasing taurine conjugated bile acids; Stabling IκBα; Inhibiting NF-κB signaling.	[169]
	*Lactobacillus salivarius* UCC118™	Modulating immune responses.	[172]
	*Lactobacillus plantarum* ZS62	Regulating oxidative stress; Modulating immune responses.	[173]

Abbreviations: a-mca: Alpha-Muricholic Acid; ß-MCA: Beta-Muricholic Acid; LCA: Lithocholic Acid; CDCA: Chenodeoxycholic Acid; UDCA: Ursodeoxycholic Acid; IκBα: Inhibitor of κB alpha.

## Data Availability

Not applicable.

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
