# Peer review of "Manipulating Microbiota in Inflammatory Bowel Disease Treatment: Clinical and Natural Product Interventions Explored"

_ijms, 2023, doi:10.3390/ijms241311004_

Round 1
Reviewer 1 Report
Good work with the review, quite comprehensive.
Author Response
Response to Reviewer 1 Comments
Point 1: Good work with the review, quite comprehensive.
Response 1: Thank you for your comments. Your encouragement and appreciation mean a lot to me. I'm thrilled that my efforts have been recognized, and your positive feedback has brought me great joy and inspiration. Once again, thank you for your support, and it will motivate me to continue striving to produce even better works.

Reviewer 2 Report
It is my pleasure to review this paper entitled “The role of the gut microbiota in Inflammatory Bowel Disease and pharmacotherapy intervention”. This is a review that summarize the role of gut microbiota in the occurrence and development of IBD and provided the latest information on the treatment of IBD by altering the structure of gut microbiota.
Overall article is well written English is fluently and adequate, title very informative about paper. The topic is very interesting. Please add strengths and limitations of the paper
Authors should enrich the paper with a recent published paper:
10.26402/jpp.2022.5.01
10.1016/j.sxmr.2021.02.002
Author Response
Thank you for your response and suggestions. We have refined the strengths and weaknesses and added them to the discussion section of our revised manuscript based on your suggestions: Firstly, we have comprehensively explored the various roles of gut microbiota in inflammatory bowel disease (IBD), including intestinal protection, immune assistance, and nutrition. This provides readers with a comprehensive understanding of the significance of gut microbiota in the pathogenesis of IBD. Secondly, the depth and breadth of the review may be constrained by length and limitations. The review covers multiple topics, but due to limited space, it may not be possible to fully explore all the details and latest research progress of each topic. Thus, The review may not cover all relevant treatment methods and drugs, as the field of IBD treatment is constantly evolving and evolving, and new drugs and treatment strategies may emerge after the article is completed.
In addition, the references you provided have been immensely helpful. I have thoroughly read the literature and incorporated relevant information into my article, as detailed in lines 29-31 added. I have made content additions in our revised manuscript based on the insights gained from the literature, as shown in below.
“IBD can induce other complications such as arthritis, cholangitis, urinary tract infections[3], and prostatitis [4]. In addition, IBD can cause stress and psychological problems in patients, resulting in decreased libido, sexual dysfunction [5], anxiety, fatigue, and even depression [6], severely impacting the quality of life of IBD patients.”

Reviewer 3 Report
According to authors, the present review aims to fill a gap regarding the role of gut microbiota and drug effects in IBD. I was able to found as much as 195 reviews only regarding the last year in PUBMED. Beside this, the review is quite caothic and not well organized, already from the title which should probably read "Effect of traditional chinese medicines and other natural products on IBD". Having said this, the tables do not reflect what is indicated in the text, nor they follow the order presented in the text. As an example table 4 is quoted at 403 line but it is shown in the Table 2. The effect of immunodepressants and antibiotics is indicated in the text to be shown in Table 1, but this is not true. The same applies to natural products effects, quoted to be shown in Table 3, but again this is not the case.
Author Response
Thanks for comments. We deeply agree with the points you have raised that there are numerous studies focusing on the correlation between gut microbes and therapeutic agents for IBD, which is a good indication that targeting microbiota is potent therapeutically strategies against IBD, it is worth for us to update and summery these studies. Specially, herbal medicine against IBD provide a novelty biochemical resources for new drug development is an important current hotspot in the field and a line of research mentioned their effects on gut microbiota. In this respect, a comprehensive summary article is urgently needed to outline these findings related to natural products and microbiota.
Regarding the title, we have accepted your suggestion to change the title to "Manipulating Microbiota in Inflammatory Bowel Disease Treatment: Clinical and Natural Product Interventions Explored". We have made modifications to the language and organization of the article. Again, I apologized for the confusion in the order of the tables due to my mistake, and I have reordered them and checked their order.
Thank you again for your valuable feedback, which has inspired me and made substantial improvements.

Round 2
Reviewer 3 Report
The paper has now greatly improved. I still found difficult to read teh tables, and I hope that the order now is correct. The title is more consistent with the text. I leave to the Editor the decision to publish or not this review, which is in my eyes not particularly interesting, according to the priorities of the Journal